# Imaging the polymerization of multivalent nanoparticles in solution

Juyeong Kim[1,2], Zihao Ou [1], Matthew R. Jones [3], Xiaohui Song[1,2] & Qian Chen [1,2,4]

Numerous mechanisms have been studied for chemical reactions to provide quantitative predictions on how atoms spatially arrange into molecules. In nanoscale colloidal systems, however, less is known about the physical rules governing their spatial organization, i.e., self-assembly, into functional materials. Here, we monitor real-time self-assembly dynamics at the single nanoparticle level, which reveal marked similarities to foundational principles of polymerization. Specifically, using the prototypical system of gold triangular nanoprisms, we show that colloidal self-assembly is analogous to polymerization in three aspects: ensemble growth statistics following models for step-growth polymerization, with nanoparticles as linkable "monomers"; bond angles determined by directional internanoparticle interactions; and product topology determined by the valency of monomeric units. Liquid-phase transmission electron microscopy imaging and theoretical modeling elucidate the nanometer-scale mechanisms for these polymer-like phenomena in nanoparticle systems. The results establish a quantitative conceptual framework for self-assembly dynamics that can aid in designing future nanoparticle-based materials.

[1] Department of Materials Science and Engineering, University of Illinois, Urbana, IL 61801, USA. [2] Frederick Seitz Materials Research Laboratory, University of Illinois, Urbana, IL 61801, USA. [3] Department of Chemistry, Rice University, Houston, TX 77005, USA. [4] Department of Chemistry, University of Illinois, Urbana, IL 61801, USA. Correspondence and requests for materials should be addressed to Q.C. (email: qchen20@illinois.edu)

A major challenge in materials chemistry and physics is to understand, predict, and control the spontaneous formation of materials from well-defined building blocks[1]. For atomic systems, chemists have developed numerous conceptual frameworks, such as reaction rate equations, transition state theory, and principles of detailed balance, to describe and quantify the organization of atoms into molecules and crystals[2–6]. For nanoscale colloidal systems, however, the governing "binding" interactions and dynamics are more difficult to model and, as a result, unified guiding principles for nanoparticle self-assembly behaviors are less common[7, 8]. A promising route to make quantitative predictions of their organization into materials with collective functions is thus to draw inspiration from models established for atomic systems.

Polymerization, the process of connecting many reactive monomers into large molecules, is a particularly suitable concept to apply to the self-assembly of colloidal nanoparticles[9–11] because in both systems: the building blocks involved are self-repeating, often with only one or two binding mechanisms; the kinetics of structure formation are governed by simple rate equations; and the building blocks react and bond directionally as a result of well-defined coordination geometries[12, 13]. The translation of such ideas to colloidal nanoparticle self-assembly has the potential to establish new rules that allow predictions regarding the formation of materials with new properties. One prominent example is on the self-assembly of nanorods into one-dimensional chains[14, 15], whose ensemble scale statistics were characterized through stationary electron microscopy snapshots and were found to follow step-growth polymerization. These studies monitored discrete states of assembled structures, not continuous, dynamic self-assembly processes, and therefore did not illustrate fundamental real-time interactions and kinetic pathways governing self-assembly.

Here, we utilize the unique capability of liquid-phase transmission electron microscopy (TEM)[16–18] to quantitatively elucidate nanoparticle polymerization by resolving the motion trajectories of individual nanoparticles in solution during self-assembly. Such single nanoparticle-level observation reveals the real-time dynamics of structure evolution, which can be quantified with a high degree of accuracy. We first confirm that in our liquid-phase TEM observations, the assemblies of our model nanoparticles have size distributions following canonical growth laws of polymerization. In addition, the valency of nanoparticle "monomers", namely their bonding geometry[19], determines assembly conditions such as the nature of attachment pathways, the energetic selection of bond angles, and the topology of the final material. This marked integration of a molecular conceptual framework with nanoscale self-assembly dynamics potentially generalizes to other materials and can help design and fabricate complex architectures of nanoparticles for desired properties and applications. The in situ liquid-phase TEM imaging, motion trajectory tracking and analysis, and interaction modeling demonstrated here can serve as a toolset potentially generalizable to reveal quantitative laws of other nanoscale self-assembly systems.

## Results

### Colloidal gold triangular nanoprisms in liquid-phase TEM. In this work, our prototypical system is the quasi-two-dimensional (quasi-2D) assembly of gold triangular nanoprisms. The prisms are geometrically sophisticated, with a high aspect ratio and a drastic variation in local curvature (ranging from flat sides to sharp tips). These features together determine the multivalent nature of individual nanoparticles, which are beyond the radially symmetric nanoparticles (nanospheres[20–24] and octapods[25])

investigated in previous liquid-phase TEM studies. In addition, the prisms also have architecture-dependent plasmonic coupling, which may be relevant to molecular diagnostics, the design of metamaterials, surface-enhanced spectroscopies, and light manipulation[26–31].

The gold triangular nanoprism building blocks used here are $90.9 \pm 9.7$ nm in side length and 7.5 nm in thickness, which are functionalized with alkyl-thiol ligands with terminal carboxylic acid functional groups (Supplementary Fig. 1)[32–35]. The strong gold-thiol bond allows the system to be prepared in deionized water with the absence of free ligands. This minimizes uneven liquid background contrast and depletion forces from ligand aggregates, which facilitates imaging and modeling of internanoparticle interactions. The ligands on the prisms are completely deprotonated into negatively charged $-COO^-$ groups ($pK_a = 3.5$–$3.7$)[33]; the particles are thus electrostatically repulsive and kept stable in solution. The prisms have large basal faces and lie flat on the liquid-phase TEM chamber substrate with freedom to move laterally in 2D while staying mostly in focus due to prism-substrate interactions[35, 36]. Their motions are slower than those predicted by Stokes-Einstein equation, likely because of an increase in solvent viscosity during imaging or the involvement of nanoparticle-substrate attractions[24, 37–40]. These effects bring the time scale of nanoparticle motions up to the temporal resolution of liquid-phase TEM instrumentation, while as we detail later, they still keep the fundamental nature of inter-nanoparticle interactions involved unchanged.

Under liquid-phase TEM, prisms first rotate and diffuse individually, but seconds after beam illumination, their self-assembly is triggered and the prisms form into linear chains via tip-to-tip connections (85% out of all prism–prism connections, Fig. 1a–c). Through radiolysis reactions with water, the imaging beam monotonically increases the ionic strength, facilitating the counter-ion screening of electrostatics, and increases the acidity, rendering a smaller charge density on prisms (estimated in Supplementary Fig. 2 and Supplementary Notes 1–2 following radiolysis equations of water[41]). Both changes reach the steady-state concentration profiles within seconds upon beam illumination[41]. They weaken electrostatic repulsion while keeping the competing van der Waals attraction the same; this promotes prism assembly. At the low-dose rates used here (10–40 e⁻ Å⁻² s⁻¹), the ligands on the prism surface have been shown to stay intact[35]. In addition, we have also shown in our previous work that at this low-dose rate range, the internanoparticle interactions and self-assembled structures modulated by the electron beam can be reproduced by implementing effective ionic strength and pH conditions outside the TEM[35]. This correlation between beam dose rates and external solution conditions enables the triggering of self-assembly during TEM imaging, which ensures capturing of the complete self-assembly dynamics starting from individual, dispersed prisms (Fig. 1a–c).

### Step-growth polymerization of colloidal monomers into chains. To quantify the resulting growth process, we relate the linear prism chains to polymer chains grown out of molecular monomers with reactive bonding sites, following step-growth polymerization[14]. In this regard, the prisms are observed to be effectively "reactive" (Fig. 1a, b) at the attractive tips; we see the prisms as "divalent" because usually two of three tips are involved in the growth into linear chains, which we attribute to steric hindrance effects between neighboring prisms (Supplementary Fig. 4). The real-time monitoring of the structural evolution allows us to track the exact size distribution of prism chains over time. Specifically, if we denote an $x$-mer as a chain comprising $x$ prisms, we see the distribution of $x$-mers shifts toward higher $x$

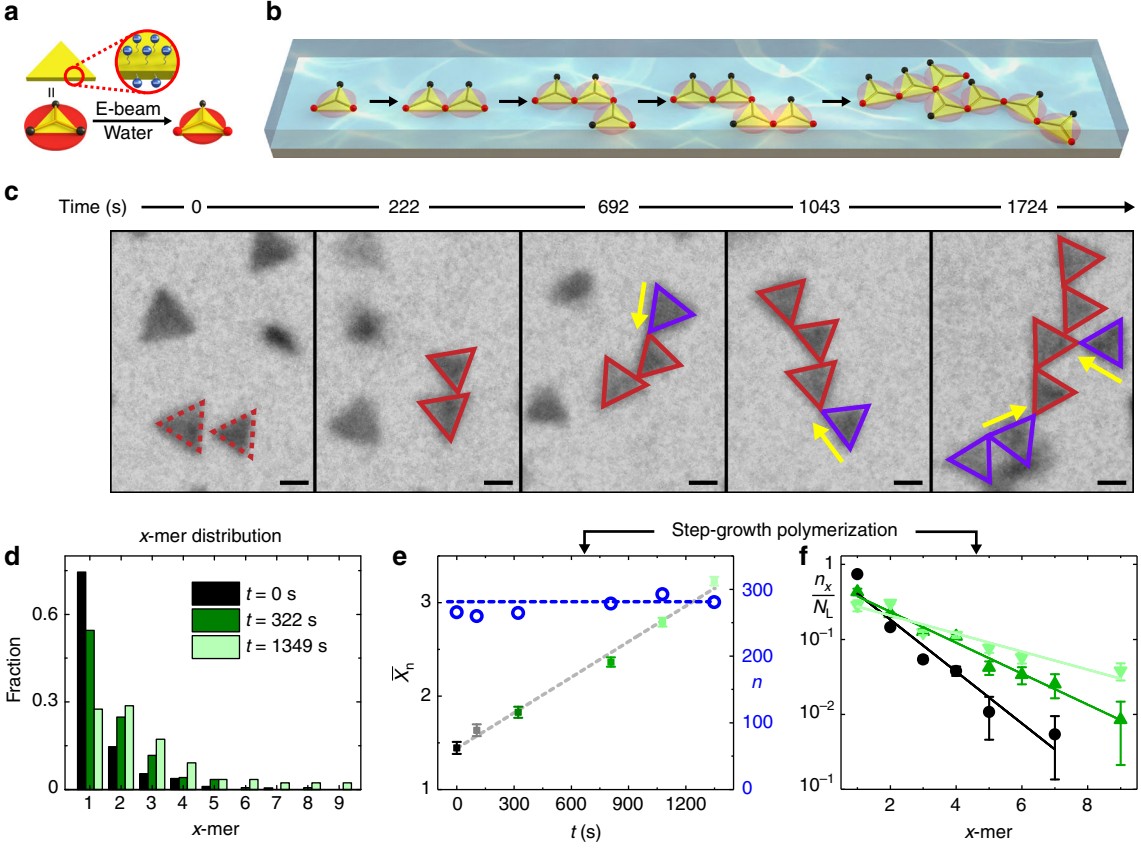

**Fig. 1** Chains assembled from gold prisms via step-growth polymerization. **a** Schematics showing a single gold triangular prism coated with negatively charged thiols and labeled with spots at the three triangular tips. The spots are *black* when the prisms are in deionized water, meaning the repulsive cloud (*red*) envelopes the prism tips and renders the prisms non-reactive. Once the prism solution is illuminated by the electron beam at the appropriate dose rates (9.8–25.8 $e^-$ Å$^{-2}$ s$^{-1}$), the prisms are rendered attachable at the tips (two spots changing to *red*) with the repulsive cloud shrunken inwards. **b** Stepwise tip-to-tip assembly schematic representing the connection scheme of the prisms highlighted in **c**. **c** Time-lapse liquid-phase TEM images showing the growth of a prism chain at a dose rate of 16.3 $e^-$ Å$^{-2}$ s$^{-1}$ (see details on image processing in Supplementary Fig. 3 and Supplementary Note 3). We overlay outlines on the prisms: *dotted red lines* for monomeric prisms before attachments, *solid red lines* for assembled prisms, and *solid purple lines* for prisms newly attached to the chain. The *yellow arrows* show the direction of prism attachment. **d** Distribution of *x*-mer, chains comprising *x* prisms, fraction changing over time (*black*: $t = 0$ s, *green*: $t = 322$ s, *light emerald*: $t = 1349$ s), which shows a shift toward higher aggregation number *x*. **e** The graph showing $\overline{X}_n$ (squares to the left *y* axis), the number-average degree of polymerization, growing linearly with time *t*, and the total number of prisms *n* (both unreacted and reacted, *blue circles* to the right *y* axis) remaining constant over time. The *gray dotted line* is the linear fitting of $\overline{X}_n - t$ relation, while the *blue dotted line* is a guide to the eye. **f** A semi-log plot showing how the fraction of *x*-mers ($n_x/N_L$) is distributed at different assembly times (*black circle*: $t = 0$ s, *light green up triangle*: $t = 808$ s, *emerald down triangle*: $t = 1078$ s). The lines are the corresponding fit based on the Flory–Schulz distribution. *Error bars* denote standard deviations from counting. *Scale bars*: 50 nm

values as time elapses, indicating an ensemble chain growth (Fig. 1d). Notice that the total number of prisms in the imaged area does not change over time (blue circles in Fig. 1e), which suggests the system is effectively closed. The number-average degree of polymerization $\overline{X}_n$ is given by $\overline{X}_n = \sum n_x x / \sum n_x$, where $n_x$ is the number of *x*-mers. As shown in Fig. 1e, $\overline{X}_n$ grows proportionally with time *t*, which is a key signature of reaction-limited step-growth polymerization. We derive an assembly rate constant *k* of $1.1 \times 10^3$ M$^{-1}$ s$^{-1}$ from the linear fitting of the $\overline{X}_n - t$ curve following $\overline{X}_n = 4[M]_o kt + 1$, where $[M]_o$ is the initial monomer concentration and *t* is the assembly time (Supplementary Note 4). This rate constant is one order of magnitude smaller than the value elicited from electron microscopy snapshots of quenched nanorod assemblies ($2.9 \times 10^4$ M$^{-1}$ s$^{-1}$)[14], probably due to slowed particle diffusion caused by prism-substrate interactions[38–40]. In addition, we characterize the change of *x*-mer fraction distribution over time, which follows the Flory–Schulz distribution, another feature of step-growth polymerization[12]. As shown in Fig. 1f, the *x*-mer fraction fits with the

relation of $n_x/N_L = (1 - p)p^{x-1}$, where $N_L$ is the total number of *x*-mers and *p* measures the extent of reaction at a specific reaction time. The extent of reaction values derived from the temporally evolving distributions increase with the assembly time (Supplementary Fig. 5a, b, Supplementary Table 1 and Supplementary Note 4) and reach at a plateau of 0.7, which suggests the equilibrated completion of the assembly. The polydispersity index (PDI) of the chains also fits well with the relation of PDI $= 2 - \frac{1}{\overline{X}_n}$, which is expected for step-growth polymerization of divalent monomers (Supplementary Fig. 5c and Supplementary Note 4).

The quantitative match of the prism assembly statistics with reaction-limited step-growth polymerization has implications potentially generalizable to other nanoparticle self-assemblies. First, the agreement of the ensemble level growth statistics with previous studies outside the TEM[14, 15] suggests that the fundamental nature of the interactions and rate laws learned in real-time liquid-phase TEM studies are consistent with self-assembly outside the TEM. Second, only a certain percentage of

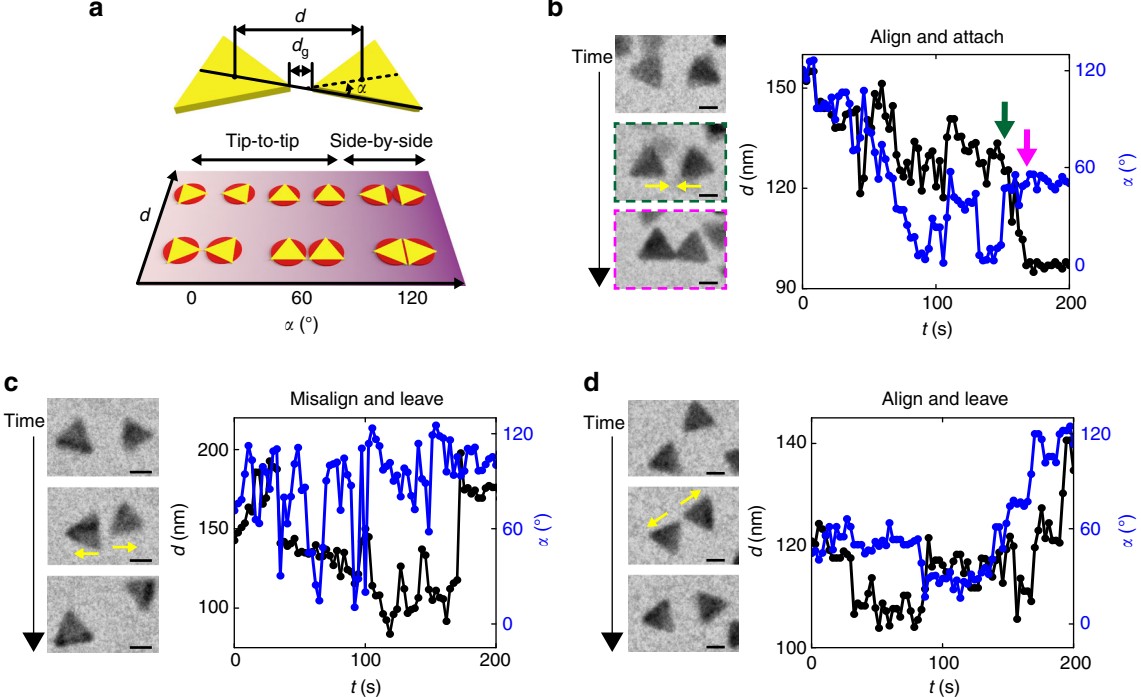

**Fig. 2** Temporal traces showing the selection of tip-to-tip attachments through long-range repulsion. **a** Schematics defining $d$, the center-to-center distance, $d_g$, the gap distance between two approaching prism tips, and $\alpha$, the relative orientation of two close prisms. The $\alpha$ angle is defined as supplementary to the angle formed by two intersecting straight lines each connecting the closest tip and midpoints of the remote prism side. Based on this definition, the angle is independent of the distance $d$. **b–d** Three representative types of time-lapse liquid-phase TEM images of two prisms approaching each other, and corresponding temporal traces characterizing changes in their center-to-center distance $d$ (*black circles*) and relative orientation $\alpha$ (*blue circles*) over time. In the case of "Align and attach" **b**, the *green arrow* in the graph corresponds to the time for the TEM image boxed in green, when the tip-to-tip orientation is selected; the *magenta arrow* in the graph corresponds to the time for the TEM image boxed in magenta, when the prisms are attached. *Scale bars*: 50 nm

two approaching particles cause a successful attachment according to the "reaction-limited" nature of self-assembly; this is reminiscent of chemical reaction processes described in the collision theory, where covalent bonds between two atoms only form when their collisions occur at suitable relative orientations[42]. Third, nanoparticles assemble following the underlying assumption of step-growth polymerization—Flory's assumption—where the reaction rate for each attachment event stays constant and independent of the growing chain length[12]. This agreement suggests that the internanoparticle interactions, the origin of "bonding" in self-assembly, are dominated by contributions from nearest neighbors, which serves as a guideline for our later pairwise internanoparticle interaction modeling.

**Tip-to-tip assembly due to long-range directional repulsion.** Going beyond the ensemble growth statistics, we next investigate what factors govern the highly preferential tip-to-tip attachments during the linear chain growth. The data we focus on is real-time rotational and translational traces of two approaching prisms. We use two orthogonal parameters to characterize the configuration of the two approaching prisms, including the center-to-center distance $d$ that measures prism positional separation, and the relative orientation $\alpha$ that measures the extent of tip-to-tip alignment (Fig. 2a). For convenience, we define an $\alpha$ value ranging from 0° to 60° as tip-to-tip alignment and an $\alpha$ value larger than 60° as the side-by-side configuration. We follow 15 temporal traces of two approaching prisms (Fig. 2b–d and Supplementary Figs. 6–7) to see how they statistically sample different outcomes when approaching each other and what approaching paths lead to successful attachments of prisms.

We group the temporal traces into three characteristic behaviors (Fig. 2b–d, Supplementary Fig. 7, Supplementary Note 5, and Supplementary Movies 1–3), all of which are consistent with a long-range repulsive effect that favors tip-to-tip attachments. "Align and attach" is the one approaching path that leads to a successful attachment, i.e., a chain growth event. In the representative trace (Fig. 2b), two prisms first start from side-by-side configuration ($\alpha = 120°$) at about 150 nm distance. As the distance shortens and then fluctuates around 120 nm, $\alpha$ narrows down to the range of 20°–60° and stabilizes at the final angle of 60° about 20 s before attachment. This observation suggests an interesting long-range effect: the tip-to-tip configuration is selected at a distance, before the prisms are in physical contact. Then the prisms are successfully attached via van der Waals attraction. The other two scenarios show approaching prisms that do not attach successfully. The representative trace for "misalign and leave" (Fig. 2c) describes two prisms occasionally diffuse into proximity side-by-side ($\alpha = 120°$). Soon after their distance is sufficiently small, about 15 nm from physical contact, the two prisms jump apart from each other, consistent with a repulsion disfavoring incorrectly oriented prisms. The only repulsion involved in this system is electrostatic repulsion. The representative trace for "align and leave" (Fig. 2d) shows that two prisms approaching at the favored tip-to-tip configuration ($\alpha$: 20°–60°) can still gradually diffuse apart afterwards. The "diffuse-apart" outcome is consistent with a long-range repulsive interaction; an otherwise attractive interaction will almost always secure successful attachments given the favored tip-to-tip configuration. The combined statistics from all the tracked traces summarize experimentally sampled inter-prism distances ($d$) and relative

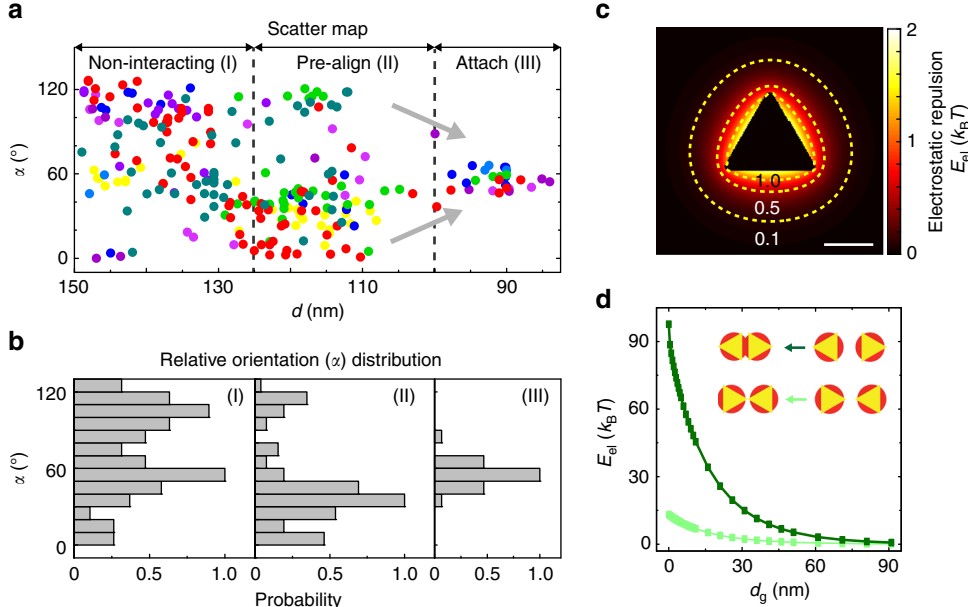

**Fig. 3** Statistical analysis of tip-to-tip attachments due to long-range repulsion. **a** A scattered map showing the distances and relative orientations sampled by two approaching prisms based on experimental liquid-phase TEM observations. Each data point color corresponds to a single trace of prism pairs. We see three regions in the map, non-interacting (I), pre-align (II), and attach (III), which correspond to distinct relative orientation distributions shown in **b**. **b** The relative orientation ($\alpha$) distributions obtained from the scattered map in **a**, changing from wide distribution (I), narrowing toward one single peak (II), to peaking at 60° for these data series (III). **c** The isoenergetic contour plots of electrostatic repulsion $E_{el}$ surrounding a single prism at 160 µM ionic strength. **d** The graph of calculated electrostatic repulsion between two approaching prisms in the side-by-side (*olive*) and tip-to-tip (*emerald*) configurations vs. gap distance $d_g$ at 160 µM ionic strength. *Scale bar*: 50 nm

orientations ($\alpha$) in the assembly coordinate (Fig. 3a). Based on the Boltzmann inversion rule, the more populated the configuration of a certain ($d,\alpha$) value, the more energetically favorable the configuration. Based on the scattered map and the distribution plot (Fig. 3a, b), the range of observed $\alpha$ values narrows as $d$ decreases, with its distribution shifting to the peaked tip-to-tip configuration. This analysis visualizes the long-range repulsive effect of electrostatic interactions directly: prisms are non-interacting when they are far away (region I), then sense a repulsive barrier and pre-align as they come closer (region II), and finally follow the tip-to-tip alignment of the lowest repulsion to attach with one another (region III).

The directional long-range repulsion of electrostatic nature has a profile determined by the shape anisotropy of prisms. Specifically, we calculate the isoenergetic contours of electrostatic repulsion for a single prism against a test charge. As shown in Fig. 3c, the contours are nearly circular, which render the repulsion the lowest at the tips and the strongest at the sides when overlaid with the triangular prism shape. Regarding the magnitude of the repulsion, we calculate how it changes as two prisms approach, in the typical tip-to-tip ($\alpha = 0°$) and side-by-side ($\alpha = 120°$) configurations respectively. As shown in Fig. 3d, the side-by-side approaching experiences a pronounced repulsion up to $100\,k_B T$, where $k_B$ is the Boltzmann constant and $T$ is the temperature, a barrier too high for particles to overcome via diffusions. In comparison, tip-to-tip aligned prisms could overcome a relatively low repulsion barrier ($15\,k_B T$) to diffuse close and fall into the range of van der Waals attraction (see more details in Supplementary Fig. 8 and Supplementary Table 2). The last step of attachment driven by this van der Waals attraction is sometimes observed as a jump from free to attached (Fig. 2b). This jump is likely due to attraction kinetics that are considerably faster than our acquisition capability[43].

Note that the long-range interaction origin of tip-to-tip attachment represents one feature of dynamics in nanoscale systems that is distinct from the micron-sized model colloids well-studied using optical microscopy. At the nanoscale, interaction ranges easily become long relative to the small nanoparticle size, which affect the dynamics of nanoparticles when they are physically apart[44–46]. Our observation is reminiscent of other long-range interaction effects where nano-sized objects start to interact at a distance away from physical contact, such as directional assembly of nanoparticles through long-range electrostatic patchiness[47].

**Bond angle selected by the nanoscale shape of prism tips.** Under the overarching tip-to-tip attachment theme, we apply the concept of bond angles in polymers to characterize the relative orientation of adjacently connected prisms. We extend the notation of $\alpha$ used earlier to the bond angle of two attached prisms. In practice, we measure the equilibrated bond angles from 84 tip-to-tip attachments in the assembled chains, which reveal a surprising bimodal distribution peaked at 0° and 60° (Fig. 4a). The bimodal distribution is also in good agreement with the two basic bonding motifs through which the individual prisms are observed to connect into chains (Fig. 4b and Supplementary Fig. 9).

High-resolution TEM imaging elucidates that the bimodal bond angle distribution is related to a seemingly trivial nanoscopic morphology detail of the prism tip. The 0° bonding motif has two prisms with rounded tips pointing at each other, while the 60° bonding motif often entails one prism with a flat tip sitting on the side of the neighboring prism (Fig. 4b, c). The existence of this hexagonal truncation in colloidally synthesized gold prisms has been commonly observed[33, 48, 49], but has barely been implicated in structural control of self-assembled materials. Note that the ligand density of truncated tips may be different from that of rounded tips[50, 51]; we show here that we do not need to include this potential difference in our interaction calculation

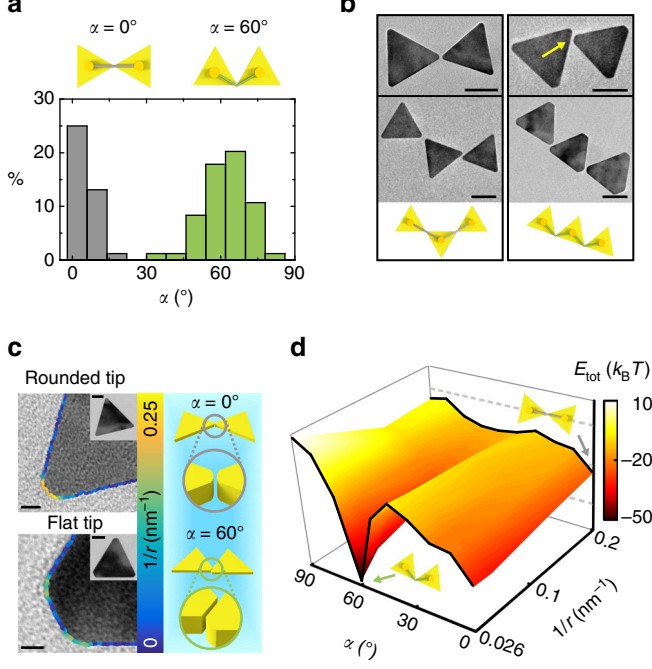

**Fig. 4** Optimal bond angle determined by prism tip curvature. **a** The bond angle ($\alpha$) distribution in the prism chains based on liquid-phase TEM observations has two peaks at 0° and 60°. The schematics show the two bonding motifs for 0° (*gray bar*) and 60° (*green bar*) bond angle connections. **b** TEM images and corresponding schematics of dimeric and trimeric chains growing out of the two bonding motifs. **c** Representative TEM images of prisms overlaid with edge contour curvature plots showing two typical curvatures ($1/r$, $r$ being the radius of the fitted circle) of the triangular prism tips. The schematics highlight the difference in tip geometry at these two bond angle configurations. **d** The dependence of total interaction energy of attached prisms ($d_g = 0.5$ nm) as a function of bond angles and tip curvatures. The two lines highlighted in *black* on the energy map show the energy curves at curvatures of 0.2 and 0.026 nm$^{-1}$, respectively. *Scale bars*: 50 nm for **b**, 5 nm for **c** and 30 nm for the inset in **c**

to reproduce the bimodal bond angle distribution. This is likely due to the negligible van der Waals attraction between organic ligands and the dominant electrostatic repulsion from the ligands on the large basal planes of prisms. As shown in Fig. 4c, the local tip curvature $1/r$ ($r$ is the radius of the circle fitting to the tip shape) is altered drastically depending on the tip morphology, from 0.2 nm$^{-1}$ for round-tipped prisms to 0.026 nm$^{-1}$ for flat-tipped prisms (see details in Supplementary Fig. 10, Supplementary Table 3 and Supplementary Note 6).

This local curvature difference modifies the energetics determining the equilibrium bond angle. To illustrate this point, we calculate the pairwise interaction of two connected prisms as a function of bond angle ($\alpha$) and prism tip curvature ($1/r$) (Fig. 4d). Here the interaction energy includes both electrostatic repulsion and van der Waals attraction, because the connected prisms are sufficiently close to experience short-range van der Waals attraction. Regarding hydration interactions that were shown relevant for the orientated attachment of bare metallic nanoparticles during their growth[43, 52], we follow established forms and calculate that the hydration interactions for a pair of prisms are constantly below 1 $k_BT$, negligible in the total free energy responsible for our observed prism attachment configurations (Supplementary Fig. 11, Supplementary Table 4 and Supplementary Note 7). As shown in Fig. 4d, the interaction energy valley propagates continuously from a stable bond angle of 0° for two round-tipped prisms (top black line) to a stable bond angle of 60°

for one round-tipped and one flat-tipped prisms (bottom black line), which agrees with the experimentally observed two bonding motifs (Supplementary Figs. 12–14). Here, the tip morphology details about 5–10 nm in length alter the interaction energy profile of nanoparticles about one order of magnitude larger in size. Physically, one can understand that one flat-tipped prism can sit parallel to one side of the other prism at 60° bond angle, thereby greatly boosting van der Waals attraction up to as much as −38 $k_BT$.

The bimodal bond angle distribution is thus attributed to the stochastic prism pairing in the suspension of round- and flat-tipped prism mixtures generically obtained from colloidal synthesis. We highlight the importance of nanoscale morphology and local surface curvature in determining the bond angle of nanoparticles, which can potentially serve as a guideline to correlate the research efforts on morphology-controlled nanoparticle synthesis with studies on their self-assembly at a high level of accuracy. In addition to this model system of triangular prisms, previous work on beveled gold prisms[53] and gold nanorods[52] has also reported how small morphology details of nanoparticles determine other interaction types, such as depletion and hydration interactions. In the nanorod paper[52], two different assembly configurations were observed and selected by contact-determined hydration interactions. In comparison, interactions of well-studied micron-sized colloidal building blocks often do not have such delicate controls, due to colloidal sizes much larger than the interaction ranges and the local structural morphology length scale[44].

**Polymer-like topology tuned by the valency of monomers**. The above understanding on how directional interactions control the chain length, self-assembly rate, and attachment geometry of linear chain assemblies motivates us to direct the prism self-assembly into different final structures. The dose rate in liquid-phase TEM serves as the handle to fine-tune the solvent conditions (Supplementary Note 1), and consequently the balance between electrostatic repulsion and van der Waals attraction. As shown in Fig. 5a, the calculated isoenergetic repulsion contour maps of a single prism have their repulsion cloud reduced as the effective ionic strength increases, which exposes more connectable tips. A slight increase to 205 µM in ionic strength is observed to allow greater steric freedom on the prisms to accommodate more than one tip-to-tip connection (Fig. 5b and Supplementary Movie 4). This increased number of bonds (valency) leads to an assembled network woven by prism nodes, i.e., branched prisms, and cyclization events (Supplementary Fig. 4 and Supplementary Note 8). The evolution of the fraction of branched prisms over time shows an obvious increase in comparison with the constant trend for linear chains (Supplementary Fig. 15). The bond angle distribution for the networks stabilizes with a narrower peak at 60° (Supplementary Fig. 15) than that for the linear chains, which shows a steric requirement to evenly distribute three connections at the node point. The prisms are more closely packed, easily falling into multiple neighbors' interaction ranges. Many-body effects, instead of pure pairwise interactions, start to act in concert to determine the bond angle and final assemblies. Both the linear and cyclic chains are otherwise hard to achieve in typical solution-phase assembly of prisms. Due to the pronounced plasmonic coupling of tip-to-tip configurations for gold triangular prisms, one often utilizes electron-beam lithography to accurately position the prisms into bow tie-like structures for local field enhancement applications[54, 55]. Our demonstration could help better understand a bottom-up approach for obtaining non-conventional, open assemblies, such as the tip-to-tip configurations here.

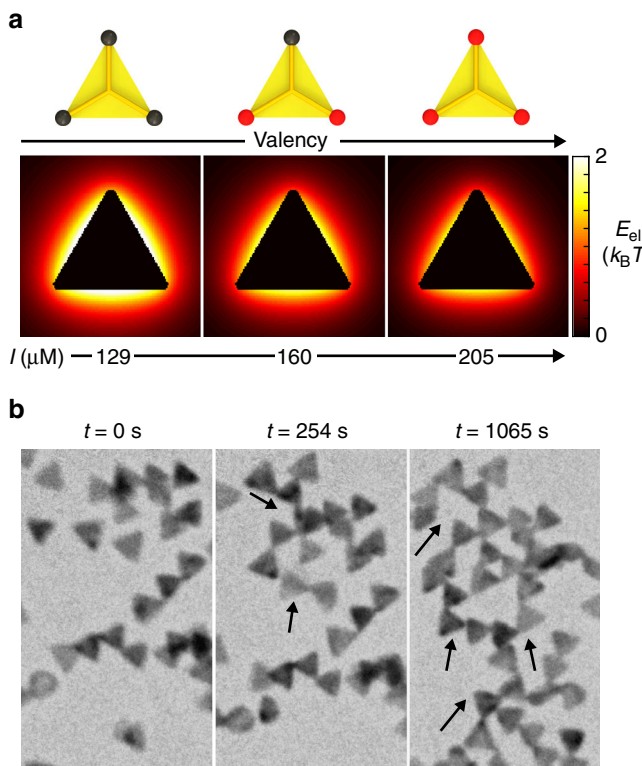

**Fig. 5** Colloidal polymer topology determined by prism coordination geometry. **a** The calculated electrostatic repulsion ($E_{el}$) maps showing that the increase in the ionic strength leads to an increase in the number of bonds, i.e., valency, of prisms (ionic strength: 129 μM, 160 μM, and 205 μM from *left* to *right*). The *red spots* in the prism schematics indicate reactive sites that determine the connection scheme and final assembly structure. **b** Time-lapse liquid-phase TEM images of the cyclic assemblies at a dose rate of 39.5 e⁻ Å⁻² s⁻¹. The *black arrows* show the direction of assembly (Supplementary Movie 4). *Scale bars*: 100 nm

## Discussion

We use liquid-phase TEM to study the fundamental dynamics and interaction details of the self-assembly behavior of anisotropic gold triangular nanoprisms. This bridging of the constituent kinetics across length scales (from molecules to nanoscale colloidal systems) can enrich the design rules and quantitative prediction for self-assembly of anisotropic nanoparticles. Combining liquid-phase TEM imaging and particle motion analysis represents a powerful toolset for determining dynamics and local morphology details in kinetic and thermodynamic studies generalizable to other nanoparticle systems. We find our theoretical modeling based on conventions of nanoparticle self-assembly explains our experimental observation, which confirms that, despite the differences of the liquid environment inside a sample chamber from conditions outside the TEM, there appears to be no fundamental difference in internanoparticle interactions. In particular, keeping liquid-phase TEM imaging at low-dose rates is crucial for avoiding irreversible particle dissolution or ligand damage[35]. Next, in the cases where the radiolysis effects have not been quantified, radical scavengers and radiation-resistant solvents can be used to mitigate potentially detrimental radiation effects[56–58]. Moreover, the quasi-2D assembly described in our prism example can be potentially achieved for solution-phase nanoparticle self-assembly in the presence of a solid substrate; there the substrate can reproduce the effects of the confinement and slowed nanoparticle diffusion based on substrate-

nanoparticle interactions outside the TEM[40, 59]. In a control experiment, we observe the tip-to-tip assembly of the same gold nanoprisms in substrate-mediated self-assembly conditions outside the TEM (Supplementary Fig. 16). A thorough, high-yield reproduction of the tip-to-tip assembly outside the TEM will involve subtle fine-tuning of parameters such as substrate-nanoparticle interactions and solvent drying speed, which is beyond the scope of this work yet promising as we demonstrate in those initial experiments.

The conceptual framework, we have built based on detailed analysis of single nanoparticle motions can potentially be generalized to a broad scope of self-assembly dynamics of nano-sized objects, including both inorganic nanoparticles and organic moieties such as polymer micelles and even proteins[60–62]. The latter two will require a greatly enhanced contrast to see the light elements otherwise invisible under the TEM as well as low-dose imaging to minimize electron beam effects. Solutions to these challenges are now starting to emerge[17, 18].

## Methods

**Chemicals used in the sample preparation**. All chemicals were used without further purification after purchase: sodium iodide (99.999%), cetyl-trimethylammonium bromide (CTAB) (BioXtra, ≥99%), gold (III) chloride tri-hydrate (≥99.9%), sodium citrate tribasic dihydrate (BioUltra, ≥99.5%), sodium borohydride (99%), L-ascorbic acid (BioXtra, ≥99.0%) and sodium hydroxide (99.99%) from Sigma-Aldrich, sodium chloride (99.3%) from Fisher Scientific, sodium phosphate monobasic monohydrate (99.0–102.0%) from EMD Millipore, sodium phosphate dibasic anhydrous (99+%) from Acros, and 2-(2-[2-(11-mer-capto-undecyloxy)-ethoxy]-ethoxy)-ethoxy-ethoxy-ethoxy-acetic acid (≥95%) from Prochimia Surfaces. All glassware was treated with aqua regia (a mixture of HCl and HNO₃ with a volume ratio of 3:1), thoroughly rinsed with water, and dried immediately before use. Nanopure water (18.2 MΩ·cm at 25 °C) purified by a Milli-Q Advantage A10 system was used in this work.

**Synthesis of gold triangular nanoprisms**. Gold triangular nanoprisms were synthesized via a seeded growth method according to literature[34, 35, 48]. First, gold nanoparticle seed solution was prepared by rapidly mixing an aqueous solution of HAuCl₄ (250 μl, 10 mM), sodium citrate (500 μl, 10 mM), and ice-cold NaBH₄ (300 μl, 10 mM) sequentially into 18.95 ml of water in a 50 ml Erlenmeyer flask and stirred at 1150 r.p.m. for 1 min. The addition of the NaBH₄ solution should be done quickly to obtain small and monodisperse gold seeds. The seed solution was incubated at 40–45 °C for 15 min before used for the growth of gold triangular nanoprisms. Then, the solution was cooled down to room temperature, and its extinction was measured using a UV–Vis spectrometer. The gold seed solution showed a plasmon resonance band at 500–505 nm with an extinction of ~ 0.26. Note that the gold seed formation specially requires thoroughly washed glassware and stir bars. Otherwise, larger and irregular seeds can often be formed.

Gold triangular nanoprisms were then grown from the as-synthesized gold seeds. Aqueous solutions of HAuCl₄ (250 μl, 10 mM), NaOH (50 μl, 100 mM), ascorbic acid (50 μl, 100 mM) and gold seeds (22 μl) were sequentially added into 9 ml of 50 mM CTAB containing 50 μM of NaI in a 20 ml scintillation vial. The solution was hand-shaken for 1 s after each addition, and the mixture was left undisturbed for 30 min. Its color gradually changed from clear to purple, indicating the formation of triangular nanoprisms along with spherical impurities. Purification of the as-synthesized product solution was performed as follows. The purple solution was transferred to a 15 ml centrifuge tube, and 0.9 ml of 2 M NaCl was added. After it was mixed well, the solution was left undisturbed for 2 h to induce face-to-face stacking of triangular nanoprisms due to depletion attraction and consequent sedimentation. It was centrifuged twice (first round: 4900 r.p.m. for 30 s and second round: 1350 r.p.m. for 5 s). After each centrifugation, supernatant was removed as much as possible using a micropipette because a tiny amount of supernatant would increase the amount of spherical impurities in the product solution. After the second round centrifugation, a couple of drops of water was first added to the sediments and then 9 ml of 50 mM CTAB was added to redisperse the product in solution. Note that direct addition of 50 mM CTAB after the second round centrifugation may cause aggregation of the product. Thus, we added water first. Successful purification yielded significant reduction of the extinction from spherical particles in UV–Vis-NIR spectroscopy.

**Thiol modification and sample solution preparation**. The surface of CTAB-bound prisms was exchanged with carboxylate-terminated thiol molecules (HS (CH₂)₁₁(OC₂H₄)₆OCH₂COOH)[33, 35]. The thiol ligands stabilize gold atoms on the surface even without an excess of free ligands in solution. In addition, the ligand exchange helped achieve better resolution for the TEM imaging, whereas the imaging of prisms in liquid could be obscured due to the high concentration of free

CTAB ligands needed for keeping prisms stable[36]. The purified prism solution in 50 mM CTAB (9 ml, extinction spectrum peak height = ~ 0.8 at 1044 nm, prism concentration = ~ 20 pM) was centrifuged twice to decrease the concentration of CTAB molecules (first round: 8800 r.p.m. for 8 min and second round: 6600 r.p.m. for 8 min). After the first round centrifugation, supernatant was removed and remaining liquid with sediments (~ 50 µl) was mixed with 8.95 ml of water. After the second round centrifugation, supernatant was removed and remaining liquid with sediments (~ 50 µl) was mixed with 3.00 ml of water. An aqueous solution of thiol molecules (44.26 µl, 7.93 mM) was added to the prism solution and incubated for 30 min. Then, it was sonicated for 5 s and 0.538 ml of 1 M pH = 8 phosphate buffer solution (PBS, composed of 0.07 M sodium phosphate monobasic monohydrate and 0.93 M sodium phosphate dibasic anhydrous) was gently added to the solution and left undisturbed overnight. The final solution contained 100 µM of thiol molecules and 0.15 M of pH = 8 PBS, where the PBS solution was present to screen the electrostatic repulsion of deprotonated thiol ligands and facilitated efficient thiol coating on the gold prism surface. The final prism concentration after the thiol modification was estimated as ~ 50 pM, assuming there was no loss of prisms during the centrifugations. During ~ 15 h of incubation, thiol-modified prisms began to assemble face-to-face stacked and form into black sediments. The solution with black sediments was used in liquid-phase TEM.

For large-scale sample check using TEM of dry samples (Supplementary Fig. 1), we decreased the ionic strength of the thiol-modified prism solution in 0.15 M pH = 8 PBS as follows. An aliquot of prism solution (5 µl) with the black sediment was transferred and mixed with 1 ml of water, which allowed the assembled prisms individually dispersed in liquid (PBS: ~ 750 µM) (Supplementary Fig. 1a for UV−Vis-NIR spectrum). Then, the diluted prism solution (1 ml) was concentrated by removing 0.75 ml of supernatant after centrifuged at 7200 r.p.m. for 2 min. The concentration of the prisms was estimated as ~ 120 pM in this solution (solution 1), which was then used for dry TEM sample preparation and substrate-mediated tip-to-tip assembly by scanning electron microscopy (SEM) as detailed below.

**Prism characterizations**. A Scinco S-4100 PDA UV−Vis spectrometer and a Varian Cary 5 G UV−Vis-NIR spectrophotometer were used for measuring plasmon resonance bands of gold seeds and prisms. A JEOL 2100 Cryo TEM and a Hitachi S-4800 SEM were used to characterize size, shape and assembly structures of prisms (Supplementary Figs. 1b, 9 and 16). For TEM, an aliquot (10 µl) of the thiol-modified prism solution (solution 1) was placed on a TEM grid that was plasma-cleaned at a low RF level for ~ 30 s using a Harrick PDC-23G basic plasma cleaner, and the specimen was dried under vacuum for ~ 15 min and then imaged.

The substrate-mediated tip-to-tip assembly was characterized using SEM. The thiol-modified prism solution (10 µl of solution 1: prism concentration = ~ 120 pM, PBS = ~ 750 µM and thiol ligands = ~ 0.5 µM) was placed on a silicon wafer (n type, 1–10 Ω · cm, thickness: 710–740 µm, Taisil Electronic Materials Corp.) and incubated in a humid condition for the nanoparticles to sediment and interact with each other before getting dried. Specifically, the humid condition was achieved by placing the silicon wafer with the sample droplet in a capped and sealed petri dish, in which two small containers (vial lids) with ~ 1 ml of water each were placed next to the silicon wafer to maintain saturated water vapor. The incubation time varied up to 6 h. After the incubation, the sample droplet was dried under vacuum for 15 min before imaging.

**Liquid-phase TEM sample preparation and imaging**. The in situ liquid-phase TEM imaging was conducted using a Hitachi 9500 TEM with a LaB$_6$ emitter at 200 kV using a liquid flow holder (Hummingbird Scientific). In a typical experiment, SiN$_x$ microchips were first plasma-cleaned at a low RF level for 20 s using a Harrick PDC-23G basic plasma cleaner. An aliquot (~ 0.1 µl) of black sediments of prisms in pH = 8 PBS was placed on a SiN$_x$ microchip (window: 50 µm × 200 µm × 50 nm, 250 nm spacer, Hummingbird Scientific) and assembled with another microchip (window: 30 µm × 650 µm × 50 nm). Deionized water was flowed through the liquid flow holder for 2 h (we tested the flow time up to 5 h and did not observe a difference) at 5 µl min$^{-1}$ to exchange the liquid environment from 0.15 M pH = 8 PBS to deionized water, which allowed the assembled prisms to be redispersed individually in liquid.

The liquid-phase TEM movies were captured by a Gatan Orius fiber-optically coupled CCD camera with an exposure time of 0.1 s. A spot size of Micro1 was used for imaging of linear prism chains, and a spot size of Micro3 was used for imaging of cyclic prism chains. The dose rate can be controlled by manipulating electron beam size, magnification, and the first condenser lens (C1). The dose rate was calculated from total pixel intensity of an acquired image with samples out following Equation (1). The conversion factor was provided as 10.19 count per e$^-$ from Gatan, Inc.

$$\text{Dose rate} = \frac{\text{Total pixel intensity}}{\text{Acquisition area} \times \text{Exposure time}} \times (1/\text{Conversion factor}) \quad (1)$$

**Calculation of electrostatic repulsion for a single prism**. This calculation concerns Fig. 3c and Fig. 5a. We consider that the prism surface is uniformly covered with negatively charged ligands (–COO$^-$). The charge density (−0.0047 C

m$^{-2}$) was calculated based on our zeta potential measurements at pH = 8 when nearly all carboxylate groups are deprotonated. In these calculations, only electrostatic repulsion was considered because side-by-side vs. tip-to-tip configurations were determined by long-range interactions. We followed previous work[36, 47, 63, 64] to model a prism as composed of two parallel charged triangular plates (the top and bottom ligand layers) positioned with the same orientation and 12.3 nm apart (the sum of a 7.5 nm prism thickness and a 4.8 nm thickness of two layer of ligands[35]). Each triangular plate was further discretized into square meshes, wherein each mesh contains one unit charge given our measured charge density. The pH value of the aqueous environment will influence the charge density on prism surface, which has been discussed in detail in Supplementary Note 2 and shown in Supplementary Table 2. To capture the electrostatic repulsion screening effect by ionic species in solution, we used Yukawa potential in our calculations[36, 63, 64]. Thus, the total electrostatic repulsion energy between one single prism at the coordinate center and a test unit charge ($e = -1.6 \times 10^{-19}$ C) on the observation plane (parallel to the charged planes and positioned at the middle of two charged triangular planes) of a vector **R** to the coordinate center is the sum of the electrostatic repulsion between each unit charge in the meshed charged planes and the test unit charge as shown in Equation (2).

$$E_{\text{el}}(\mathbf{R}) = \sum_{\mathbf{R}_i} \frac{eZ}{4\pi\varepsilon\varepsilon_0 |\mathbf{R} - \mathbf{R}_i|} e^{-\frac{|\mathbf{R}-\mathbf{R}_i|}{\lambda_D}} \quad (2)$$

The sum over $\mathbf{R}_i$ (the coordination vector of unit charge $i$ in the meshed charged planes) samples all the charged ligand positions on the prisms. The Z value is the effective charge of the ligand. The $\varepsilon$ value is the relative dielectric constant of the liquid, and the $\varepsilon_0$ value is the permittivity of vacuum. The $\lambda_D$ value is the Debye screening length of the solution determined only by the total ionic strength $I$ in our system. Debye length was calculated by $\lambda_D = [\varepsilon\varepsilon_0 RT/(2IF^2)]^{\frac{1}{2}}$, in which $R$ is the gas constant and $F$ is the Faraday constant.

According to the above model, we presented the electrostatic repulsion energy profile surrounding a single prism in the liquid condition for the linear chain growth (Fig. 3c). The repulsion zone becomes smaller when the ionic strength increases (Fig. 5a), and this can influence the prism valency and their assembly structure.

**Calculation of electrostatic repulsion for a pair of prisms**. This calculation concerns Fig. 3d on how the electrostatic repulsion energy between a pair of approaching prisms change at different relative orientation ($\alpha$). The electrostatic repulsion energy between two approaching prisms was calculated similarly to the above, by summing over a pairwise Yukawa potential between all the negatively charged ligands of two approaching prisms. We used the same discretized model of prisms as above and the total electrostatic repulsion energy between two prisms is given by Equation (3):

$$E_{\text{el}} = \sum_{\mathbf{R}_i} \sum_{\mathbf{R}_j} \frac{Z_i Z_j}{4\pi\varepsilon\varepsilon_0 |\mathbf{R}_i - \mathbf{R}_j|} e^{-\frac{|\mathbf{R}_i-\mathbf{R}_j|}{\lambda_D}} \quad (3)$$

Here $i$, $j$ are the $i$th unit charge on one prism and the $j$th unit charge on the other prism. We included all pairs of interactions when calculating the mutual electrostatic interaction without a cutoff distance because the total ionic strength in our system is low, which gives a relatively large Debye length and renders a long-range electrostatic interaction. To quantitatively investigate how the repulsion interaction energy varies for two prisms approaching at different approaching orientations, we chose two representative relative orientations: $\alpha = 0°$ and $\alpha = 120°$, and studied the repulsion energy vs. the gap distance ($d_g$). Details of the definition on bond angle and gap distance are given in Fig. 2a. The calculation model was initialized with the geometry of two prisms at a fixed bond angle and then by gradually shifting one prism positionally at the fixed bond angle to change the gap distance. In Fig. 3d, the electrostatic repulsion energy is shown to be significantly smaller at the tip-to-tip configuration ($\alpha = 0°$) than the side-by-side configuration ($\alpha = 120°$).

**Van der Waals attraction energy between a pair of prisms**. Van der Waals attraction energy is widely used to explain the assembled structure of nanoparticles and microparticles[47, 64, 65]. Most studies with spherical particles or planar geometry used established forms for the van der Waals energy calculations[35, 66–68], whereas an exact formula for anisotropic particles has not been defined and several other methods have been used to overcome the difficulties[47, 65].

We followed previous literature and used an atomic scale discrete model[65] to calculate the van der Waals attraction energy between a pair of prisms where the prisms are discretized into spheres with a diameter of $\sigma = 0.332$ nm (diameter of a gold atom). This method can easily account for shape details of prism tips in our experiments. Then, the van der Waals attraction energy between two prisms was calculated by summing up the interaction between all pairs of atoms in the two prisms. The total van der Waals energy

between two prisms was given by Equation (4).

$$E_{vdw} = \sum_{\mathbf{R}_i} \sum_{\mathbf{R}_j} -\frac{H\sigma^6}{\pi^2 |\mathbf{R}_i - \mathbf{R}_j|^6} \qquad (4)$$

The sum over $\mathbf{R}_i$ and $\mathbf{R}_j$ goes through all the atoms in one prism and the atoms in the other prism. The $H$ value is the Hamaker constant, which considers the gold –gold interaction via water $(10^{-19} \text{ J})^{69}$. Previous studies have shown that the pairwise van der Waals interaction energy is mostly contributed by the nearest contacting regions. To balance the computation accuracy and speed, a cutoff value of 15 nm atom–atom distance is chosen in the pairwise van der Waals energy calculation. This cutoff value could greatly reduce the calculation time and the calculation error has been tested to be less than 0.1% compared with the full atomic scale model. Combining the electrostatic repulsion and van der Waals attraction together, the total interaction energy was calculated to explain the triggering of self-assembly of prisms by increased ionic strength and a lowered electrostatic repulsion barrier (Supplementary Fig. 8).

**Total interaction map for a pair of connected prisms**. This calculation concerns Fig. 4d and Supplementary Figs. 13–14 on how the total interaction energy of two connected prisms changes over the bond angle and local curvature of prism tips. Our measurements from 100 prism tips show two populations of the tip curvature (Supplementary Fig. 10). We divided them into two types of tip geometry, rounded and flat tips. All parameters used in these calculations are summarized in Supplementary Table 3. The gap distance $d_g$ between two prisms is also measured using high magnification liquid-phase TEM images of assembled chains and the distribution of $d_g$ is shown in the inset of Supplementary Fig. 11, with a range of 0.5–2.4 nm. This range of gap distance is smaller than the thickness of two layers of surface ligands $(4.8 \text{ nm})^{35}$ because the ligands do not form into closely packed monolayers at the prism tips$^{47}$. We chose the typical value of $d_g = 0.5$ nm in the following calculations, and the exact gap distance value does not change the qualitative trend of bond angle selection. Detailed interaction energy calculations of Fig. 4d were carried by changing bond angle $\alpha$ and local curvature of the tips $(1/r)$ with a fixed gap distance $d_g$ to study what bond angle are favored at a certain local curvature of the tips. The configuration of the two prisms can be determined by $\alpha$ and $d_g$ only, and another parameter (rotation center) was introduced to sample the local energy minimum. For each $\alpha$ value, we sampled six different rotation centers (Supplementary Fig. 12) and calculated van der Waals attraction and electrostatic repulsion energy for each geometry following Equations (3) and (4). The rotation center which gave the smallest energy value was chosen for total interaction energy at each bond angle $\alpha$ in Fig. 4d. Following the previous models, the van der Waals attraction and electrostatic repulsion energy can be calculated separately for two prisms of varying tip curvature. We fixed the tip width of one prism to be 5.6 nm (rounded tip width) and tip radius of 5.0 nm (rounded tip radius). The other prism was built with a fixed tip width of 9.3 nm (flat tip) and the tip radius was varied from 5.0 nm (radius of rounded tip) to 38 nm (radius of flat tip).

We first considered one extreme case of two rounded tips at a given rotation center with the center between two prism tips at $\alpha = 0°$ (Supplementary Fig. 13). Electrostatic repulsion energy increases monotonically with the bond angle $\alpha$. The total interaction energy was the sum of van der Waals attraction and electrostatic repulsion energy, and a local total energy minimum was centered at $\alpha = 0°$ (Supplementary Fig. 13a). Similarly, calculations were carried out at different ionic strengths, and the local minimum position did not change (Supplementary Fig. 13b). We also considered the other extreme case of a prism with one flat tip rotating around the other round-tipped prism. They were studied for a given rotation center which samples the lowest energy at $\alpha = 60°$ (Supplementary Fig. 14). Electrostatic repulsion energy still increases monotonically with bond angle, but van der Waals attraction energy shows a local energy minimum at $\alpha = 60°$ (Supplementary Fig. 14a). This local minimum of total interaction energy at $\alpha = 60°$ exists over the span of different ionic strengths (Supplementary Fig. 14b).

**Data Availability**. All the other remaining data are available within the article and Supplementary Files, or available from the authors upon request.

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

## Acknowledgements

This work was supported by the U.S. Department of Energy, Office of Basic Energy Sciences, Division of Materials Sciences and Engineering under award no. DE-FG02-07ER46471, through the Frederick Seitz Materials Research Laboratory at the University of Illinois.

## Author contributions

J.K., M.R.J., and Q.C. designed the experiments. J.K., Q.C., and X.S. performed the experiments. Z.O. contributed to the theoretical modeling of interparticle interaction and energy diagrams. All contributed to the analysis and understanding of the work and the writing of the manuscript.

## Additional information

**Competing interests:** The authors declare no competing financial interests.

