## [Peer Review File · Nature Communications]

Reviewers' comments:

Reviewer #1 (Remarks to the Author):

Summary: This paper describes a liquid-phase TEM methodology to observe the self-assembly behavior of gold triangular nanoprisms. The carboxylic acid functional groups on the prisms go from a “noninteracting” to “interacting” state under exposure to electron radiation, allowing for real-time direct imaging of self-assembly behavior. The nanoprisms were observed to follow 2D step-growth polymerization, where specific tip-to-tip attachments were found to have optimal “bond angles” due to long-range repulsion between sides of prisms, tip curvature, and activation of corners.

Recommendation: Publish

Specific Comments: I consider this manuscript to be a very thorough investigation of the self-assembly of nanoprisms. All of the claims are supported with significant and specific data, and the analysis is in-depth and comprehensive for the building block presented. Drawing analogues to atomic polymerization is very compelling and will not only appeal to the wider audience of Nature Communications, but also provide an interesting angle in which the work can be understood by a broad group of people. However, the bulk of the manuscript reads as an investigation of this specific building block's (triangular nanoprism) assembly behavior. In the authors' own words, this data and analysis is “potentially generalizable” to other systems and building blocks, but it is not entirely clear how true this statement is, given the data that is presented. As this is the only strong critique of this manuscript as written, I feel that it would be worthy of publication in Nature Communications, as is, but I offer the following comments to the authors for improving this paper.

1. The figures as presented (particularly 2 and 3) are very busy and do not quickly and easily convey the intended message. I recommend simplifying the message of each to make the overall implication of the figure more readily accessible.
2. I believe that presenting this work as a comprehensive and novel method to utilize the unique abilities of liquid-phase TEM to observe and quantitatively analyze colloidal self-assembly on the nanoscopic scale would make the work more generalizable, more accessible/interesting to a wider audience, and overall stronger. I encourage the authors to adjust the manuscript to focus more heavily to analysis methods/strategy and present nanoprisms as an ideal “model system” to show the effectiveness of the technique in drawing parallels between colloidal self-assembly and atomic polymerization.

Reviewer #2 (Remarks to the Author):

This manuscript describes ensemble and single particle scale dynamics of nanoparticle self-assembly, interpreting the dynamics and kinetics with a polymerization framework. The single particle scale observations are intriguing, but the ensemble scale comparisons of self-assembly to polymerization reveal no new physics. Secondly, the authors do not sufficiently acknowledge and cite previous related work in the field concerning polymerization models for self-assembly and directional interparticle interactions, both of which have been previously investigated. I still believe the manuscript would find broad interest within the nanoscience community and is sufficiently novel for publication in nature

communications. I recommend publication only after a major revision. Detailed comments are listed below.

1.) In the abstract and description of results in Figure 1, the authors all but ignore previous work on applying polymerization models to interpret nanoparticle assembly kinetics. The applicability of polymerization models to self-assembly was initially established by the Kumacheva group in 2010. (see Liu et al, Science, 329, 2010) The authors hang a lot of the novelty of this manuscript on drawing this parallel between assembly and polymerization as a new concept, when in reality this parallel has been known for a while. The authors should acknowledge this up front in the introduction and compare their results to this previous paper.

2.) In general, the dynamics the authors are observing are likely very different than self-assembly in macroscopic systems due to different transport and diffusion rates, confinement, solid interfaces, and beam initiated processes. The authors should comment on how liquid cell dynamics could be different from those in unconfined systems.

3.) I don't believe that the quantitative repulsive forces the authors measure are good representations of what would occur in unconfined systems. The thermal energy kT is not the natural scale for this system, because diffusion does not occur due to Brownian motion in this case, it is a surface mediated process (see Chee et al., J. Phys Chem C, 2016; Verch et al., Langmuir, 2015; Woehl and Prozorov, J. Phys Chem C, 2015; Powers et al, Nano Letters, 2017). The natural energy scaling for this system is somehow related to the energy transfer from the electron beam to the nanotriangles, however, this scale probably cannot be easily determined.

4.) The authors indicate that only electrostatics and van der Waals control assembly in this case. Several recent reports have suggested the influence of solvation forces are significant in mediating self-assembly (Welch et al., ACS Nano, 10, 2015; Anand et al., Nano Letters, 16, 2016). But the authors have not included these effects in their interaction potential energy model. There are standard models for solvation forces, the authors should test these as well in their interaction model.

5.) The results showing bimodal bonding distributions is very similar to the work of Welch et al cited above, and thus again not an entirely new concept. Welch et al also showed two predominant bonding angles for oriented attachment of silver nanorods, which the authors interpreted in terms of differences in solvation shell at various facets on the ends of the nanorods. This result is somewhat similar to what the authors are proposing here, as the bonding depended on facet geometry. I would like to see comparison to this previous article as well, as there are a number of similar aspects between the two studies.

6.) There seems to be an error in the second paragraph of page 10, the authors refer to figure 3e, which does not exist.

Reviewer #3 (Remarks to the Author):

The manuscript 'Imaging the Polymerization of Multivalent Nanoparticles in Solution' is an elegant piece of work which should be published after some revisions. While the work is novel and very interesting, at least from my point of view, I wonder if it would have the same appeal for the broad readership of Nature Communication.

More generally, one result that is particularly intriguing in my opinion is the fact that the 'dance' of nanoparticles in the align and bind case does not seem to follow a purely stochastic process. While the authors carefully described the mechanisms leading to the alignment and attachment of the particles, I would be interested in a more detailed description of what is happening, what are the driving forces, during the last step of the attachment the 'jump' from free to attached.

Furthermore, in some previous research endeavors (ref 34), the authors used similar nanoplatelets and observed the formation of 'meta-rods'. Could the authors briefly comment on how the same particles with the same ligand, in very similar conditions self-assembled in such different structures.

On page 3, the authors mentioned that the self-assembly is triggered by a combination of change in pH and ionic strength in the cell and they offer in the SI a calculation of the change of both pH and I as function of e- dose. Given that this is the driving force leading to the self-assembly, it would be interesting to either measure those value on chip during the self-assembly process or alternatively to comment on how pH and I change with time during the reaction. If I understood correctly the self-assembly is occurring in the liquid cell in absence of flow and one could expect a build-up of I and pH over time, given that the authors discuss the kinetic of self-assembly, this raises the question on how the self-assembly process is affected by those changing reaction conditions. More simplistically, how does the time-scale of the pH and I change compares to the time-scale of self-assembly?

On page 5, the authors discussed the fact that the prisms acted as divalent species and referred to fig S4 for more details. While the statistics in the SI indicated that in the linear assemblies the divalent species are favored compared to the trivalent species, this is not such a clear case for the cyclic chains. To get a better feel for the real fraction of divalent to trivalent species, it would be interesting to know what are the fraction of linear to cyclic chains in the sample, and more over what is the overall fractions of monovalent, divalent, trivalent and multivalent species in the entire system and more over how these fractions are changing when the conversion increases. How does branching vary with conversion?

On page 5, the authors claimed that the self-assembly follows a step-growth kinetic. It would be interesting to see this claim supported not only with X_n vs t , but also with the change in polydispersity (X_w/X_n) over time.

On page 8, the authors discussed how certain bond angles are more favored than others (fig2). In the supporting information the authors also both measure the experimental distribution of bond angle (fig S12) and develop a model to calculate the energy associated with each configuration (Fig S11). Could the authors directly compared the 'theoretical' results with the experimental results, to get a better sense of how accurate the phenomenological description given of this complex self-assembly mechanism is accurate?

On page 10, the authors discussed the effect of the tip curvature on the more favored self-assembled structure. Could the authors comments on how the change in the prism shape might influence the crystalline facet exposed and how this could lead to different grafting density of the ligand on the surface which in turn might influence the self-assembly process.

On page 13, the authors give some experimental details. However, it is not enough to enable other to reproduce these experiments. They do not report what is the final concentration of the gold nanoplatelets suspension, nor do they describe with enough details the ligand exchange procedure and cleaning process used to make sure that there is no remaining CTAB in solution.

On page 13, the authors described the model used for their calculation. Furthermore, some parameter

used in the calculations are given in the SI. However, it would be interesting to see a more detailed description of the modelistic approach used maybe in SI.

Finally, on a cosmetic note, the authors seem to favor (fig 1d, 1f, 2f) a color scheme based on the use of light green and slightly darker green, the difference is not obvious.

Reviewer 1

Reviewer 1 considered our work as thorough and appealing to the broad audience, with all the claims supported with significant specific data and in-depth analysis. The reviewer recommended publishing as it is, and at the same time, offered great suggestions for us to improve, which we have followed as listed below.

Reviewer 1 Suggestion 1: *The figures as presented (particularly 2 and 3) are very busy and do not quickly and easily convey the intended message. I recommend simplifying the message of each to make the overall implication of the figure more readily accessible.*

Reply: We thank the reviewer for this great comment! In the revised manuscript, we simplified the figures and making them more readily accessible. In particular, we made the following changes.

- (i) We divided the original Figure 2 into two figures (Figures 2 and 3). The revised Figure 2 focuses solely on the temporal traces of two approaching prisms, and the revised Figure 3 focuses on statistical analysis of the interactions of two approaching prisms.
- (ii) We reorganized the original Figure 3 to make it less heavily loaded with texts and more accessible.

Reviewer 1 Suggestion 2: *I believe that presenting this work as a comprehensive and novel method to utilize the unique abilities of liquid-phase TEM to observe and quantitatively analyze colloidal self-assembly on the nanoscopic scale would make the work more generalizable, more accessible/interesting to a wider audience, and overall stronger. I encourage the authors to adjust the manuscript to focus more heavily to analysis methods/strategy and present nanoprisms as an ideal “model system” to show the effectiveness of the technique in drawing parallels between colloidal self-assembly and atomic polymerization.*

Reply: We thank the reviewer for the great suggestion! In the revised version, we have made changes throughout the text to articulate more how the methodology can be generalizable to other nanoparticle systems. Here we highlight a few large changes:

- (i) In the introduction, we now articulate upfront, “This marked integration of molecular conceptual framework into nanoscale self-assembly dynamics potentially generalizes to other nanoparticles and can help design and fabricate complex architectures of nanoparticles for desired properties and applications. The in-situ liquid-phase TEM imaging, motion trajectory tracking and analysis, and interaction modeling demonstrated here can serve as a toolset potentially generalizable to reveal quantitative laws of other nanoscale self-assembly systems.” We also moved the detailed description on the model system of prisms from the last paragraph of introduction to the “Results” section to keep the introduction more generic about nanoparticle self-assembly.
- (ii) For each subsection under “Results”, we have one paragraph at the end of the subsection highlighting how the key physical concepts can be generalizable to other nanoparticle systems.
- (iii) In the “Discussion” section, we added extended discussions and additional experimental results on one key question regarding the long-term impact of liquid-phase TEM studies of nanoparticle self-assembly: how to relate in-situ observations with practical wet chemistry conditions outside TEM:

“Combining liquid-phase TEM imaging and particle motion analysis represents a powerful toolset for determining dynamics and local morphology details in kinetic and thermodynamic studies generalizable to other nanoparticle systems. We find our theoretical modeling based on conventions of nanoparticle self-assembly explains our experimental observation, which confirms that, despite the differences of the liquid environment inside a sample chamber from conditions outside the TEM, there appears to be no fundamental difference in internanoparticle interactions.

In particular, keeping liquid-phase TEM imaging at low doses is crucial for avoiding irreversible particle dissolution or ligand damage.³⁵ Next, in the cases where the radiolysis effects have not been quantified, radical scavengers and radiation-resistant solvents can be used to mitigate potentially detrimental radiation effects.⁵⁶⁻⁵⁸ Moreover, the relevance of the quasi-2D assembly described in our prism example is not limited to liquid-phase TEM studies. Solution-phase nanoparticle self-assembly in the presence of a solid substrate has also been used to achieve a similar quasi-2D assembly scheme and to reproduce these effects of the confinement and decrease of nanoparticle diffusion due to substrate-nanoparticle interactions outside the TEM.^{40,59} In a control experiment, we show that the same gold nanoprisms can also assemble tip-to-tip in substrate-mediated self-assembly outside the TEM (Supplementary Fig. 15). We observed the two representative bond angles consistent with the local curvature argument in Fig. 4. A thorough reproduction of the tip-to-tip assembly outside the TEM in high yield will involve subtle fine-tuning of parameters such as nanoparticle-substrate interactions (responsible for both slowed diffusions and confined environments), which is beyond the scope of this work. However, these initial data indicate that such conditions exist and allow access to tip-to-tip assembly outside the TEM.”

Reviewer 2

Reviewer 2 considers the manuscript would find broad interest within the nanoscience community and is sufficiently novel for publication in Nature Communications. The reviewer recommended publication after following the detailed comments, which we have addressed as listed below.

Reviewer 2 Comment 1: *The single particle scale observations are intriguing, but the ensemble scale comparisons of self-assembly to polymerization reveal no new physics.*

Reply: We thank the reviewer for the comment! We would like to note that we only discussed in Figure 1 about the ensemble scale comparison based on step-growth polymerization. We presented this ensemble scale comparison at the beginning, to show our liquid-phase TEM observations are consistent with previous self-assembly studies outside TEM. However, Figures 2-5, the focus of this manuscript, are all based on systematic observations of single particle scale dynamics in time, which as the reviewer commented, have not been probed in previous polymer-schemed self-assembly work. The fundamental understanding associated with Figures 2-5, such as long-range repulsion mediated tip-to-tip assembly, local curvature-mediated bond angle selection, and in-situ modulation of the “polymer” architectures via dynamically changing the valency of building blocks, are distinct from previous studies based on stationary EM snapshots. In the revised draft, we have clarified this point with additional text (see our reply to the comment below).

In the abstract and description of results in Figure 1, the authors all but ignore previous work on applying polymerization models to interpret nanoparticle assembly kinetics. The applicability of polymerization models to self-assembly was initially established by the Kumacheva group in 2010. (see Liu et al, Science, 329, 2010) The authors hang a lot of the novelty of this manuscript on drawing this parallel between assembly and polymerization as a new concept, when in reality this parallel has been known for a while. The authors should acknowledge this up front in the introduction and compare their results to this previous paper.

Reply: We thank the reviewer for the comment! We agree with the reviewer that the applicability of polymerization models to self-assembly was initially established by the Kumacheva group in 2010 and it would be great to compare our results with that in the previous paper. In the original manuscript, we have cited this paper and made a quantitative comparison in Figure 1, “This rate constant is one order

of magnitude smaller than the value elicited from electron microscopy snapshots of quenched nanorod assemblies ($2.9 \times 10^4 \text{ M}^{-1}\cdot\text{s}^{-1}$),¹⁴ probably due to the prism-substrate interactions that have been reported to slow down particle diffusions.³⁹⁻⁴¹”

We could not compare our results in Figures 2-5 with this previous work because Figures 2-5 were based on single nanoparticle dynamics studies. In the revised draft, to follow the reviewer’s suggestion to acknowledge this previous work upfront and to make the additional single nanoparticle dynamics aspect more obvious, we added in the introduction:

“One prominent example is on the self-assembly of nanorods into one-dimensional chains,^{14,15} whose ensemble scale statistics were characterized through stationary electron microscopy snapshots and were found to follow step-growth polymerization. These studies followed discrete states of assembled structures, not continuous, dynamic self-assembly processes, and therefore did not illustrate fundamental real-time interactions and kinetic pathways governing the self-assembly. To elucidate the polymerization analogy quantitatively at the single nanoparticle dynamics level, here we utilize the unique capability of liquid-phase transmission electron microscopy (TEM)¹⁶⁻¹⁸ to resolve the motion trajectories of individual nanoparticles in solution during self-assembly.”,

and added more comparisons in the text for Figure 1 with previous work by Kumacheva:

“First, the agreement of the ensemble level growth statistics with previous studies outside the TEM^{14,15} suggests that the fundamental nature of the interactions and rate laws learned in real-time liquid-phase TEM studies are consistent with self-assembly outside the TEM.”

Reviewer 2 Comment 2: *In general, the dynamics the authors are observing are likely very different than self-assembly in macroscopic systems due to different transport and diffusion rates, confinement, solid interfaces, and beam initiated processes. The authors should comment on how liquid cell dynamics could be different from those in unconfined systems.*

Reply: We thank the reviewer for this great and insightful comment! We completely agree with the reviewer that it is important to acknowledge the differences of self-assembly in liquid-cell environment and in macroscopic systems, and potentially to transfer the knowledge learned in liquid-phase TEM studies to experiments outside TEM. In the revised draft, we made the following revisions throughout the text to better articulate the differences and correlations between liquid-phase TEM and other self-assembly studies:

- (1) In the second paragraph of “Results”, where we first discussed nanoparticle diffusion, we added a more thorough explanation and citations on how NPs diffuse differently within liquid-phase TEM chambers:

“Their motions are slower than that predicted by Stokes-Einstein equation, likely because of an increase in solvent viscosity during imaging or the involvement of nanoparticle-substrate attractions.^{24,37-40} These effects bring the time scale of nanoparticle motions up to the temporal resolution of liquid-phase TEM instrumentation, while as we detail later, they still keep the fundamental nature of internanoparticle interactions involved unchanged.”

- (2) In the third paragraph of “Results”, where we first discussed beam-induced effects on nanoparticle self-assembly, we reorganized the text to highlight the predictable beam effect at low dose rates, and our previous work (ref 35) where we reproduced beam-induced superlattice structure change in liquid-phase TEM by increasing ionic strength outside TEM; in that work, we calibrated beam effects on self-assembly by a quantitative correlation of liquid-phase TEM and small-angle X-ray scattering (SAXS, a non-invasive technique commonly used to study solution-phase nanoparticle self-assembly):

“At the low dose rates used here ($10 - 40 \text{ e}^{-}\text{\AA}^{-2}\text{s}^{-1}$), the ligands on the prism surface have been shown to stay intact.³⁵ In addition, we have also shown in our previous work that at this low dose rate range the internanoparticle interactions and self-assembled structures modulated by electron beam can be reproduced by implementing effective ionic strength and pH conditions outside the TEM.³⁵ This correlation between beam dose rates and external solution conditions enables the triggering of self-assembly during TEM imaging, which ensures capturing of the complete self-assembly dynamics starting from individual, dispersed prisms (Fig. 1a-c).”;

- (3) In the discussion for Figure 1, we highlighted the consistency in step-growth polymerization statistics between liquid-phase TEM and studies outside TEM, “First, the agreement of the ensemble level growth statistics with previous studies outside the TEM^{14,15} suggests that the fundamental nature of the interactions and rate laws learned in real-time liquid-phase TEM studies are consistent with self-assembly outside the TEM.”;
- (4) In the “Discussion” section, we also added discussions and citations on how self-assembly studies in liquid-phase TEM can be related to experiments outside TEM by working at sufficiently low dose rates and using radical scavengers that mitigate the beam effects:

“Combining liquid-phase TEM imaging and particle motion analysis represents a powerful toolset for determining dynamics and local morphology details in kinetic and thermodynamic studies generalizable to other nanoparticle systems. We find our theoretical modeling based on conventions of nanoparticle self-assembly explains our experimental observation, which confirms that, despite the differences of the liquid environment inside a sample chamber from conditions outside the TEM, there appears to be no fundamental difference in internanoparticle interactions. In particular, keeping liquid-phase TEM imaging at low doses is crucial for avoiding irreversible particle dissolution or ligand damage.³⁵ Next, in the cases where the radiolysis effects have not been quantified, radical scavengers and radiation-resistant solvents can be used to mitigate potentially detrimental radiation effects.⁵⁶⁻⁵⁸”;

furthermore, we discussed potential paths to reproduce the effects of confinement and decreased nanoparticle diffusion due to substrate-nanoparticle interactions outside TEM:

“Moreover, the relevance of the quasi-2D assembly described in our prism example is not limited to liquid-phase TEM studies. Solution-phase nanoparticle self-assembly in the presence of a solid substrate has also been used to achieve a similar quasi-2D assembly scheme and to reproduce these effects of the confinement and decrease of nanoparticle diffusion due to substrate-nanoparticle interactions outside the TEM.^{40,59}”

- (5) We performed new experiments (Supplementary Fig. 15) to show the same gold nanoprisms are also observed to assemble tip-to-tip using substrate-mediated self-assembly outside TEM:

“In a control experiment, we show that the same gold nanoprisms can also assemble tip-to-tip in substrate-mediated self-assembly outside the TEM (Supplementary Fig. 15). We observed the two representative bond angles consistent with the local curvature argument in Fig. 4. A thorough reproduction of the tip-to-tip assembly outside the TEM in high yield will involve subtle fine-tuning of parameters such as nanoparticle-substrate interactions (responsible for both slowed diffusions and confined environments), which is beyond the scope of this work. However, these initial data indicate that such conditions exist and allow access to tip-to-tip assembly outside the TEM.”

We also hope to note that the connections are still, as the reviewer pointed out, under development. This is something essentially unavoidable when applying an emergent technique to a new question. As we noted in the conclusion, more experiments are needed to have a more conclusive comparison.

Overall, our emphasis is that by revealing, quantifying, and compensating for differences, one can correlate liquid-phase TEM studies with experimental conditions outside TEM and generate guidance on self-assembly with information not accessible with other means. This is something we demonstrated in our previous work (ref 35), and which are now articulated more thoroughly in the revised draft.

Reviewer 2 Comment 3: *I don't believe that the quantitative repulsive forces the authors measure are good representations of what would occur in unconfined systems. The thermal energy kT is not the natural scale for this system, because diffusion does not occur due to Brownian motion in this case, it is a surface mediated process (see Chee et al., J. Phys Chem C, 2016; Verch et al., Langmuir, 2015; Woehl and Prozorov, J. Phys Chem C, 2015; Powers et al., Nano Letters, 2017). The natural energy scaling for this system is somehow related to the energy transfer from the electron beam to the nanotriangles, however, this scale probably cannot be easily determined.*

Reply: We thank the reviewer for this thoughtful comment! We completely agree with the reviewer that the natural energy scaling is uncertain due to the still undecided energy transfer from the beam to individual nanoparticles (previous reports have shown consistently that heating is negligible though, see Zheng et al., Nano Lett. 2009, 9, 2460; White et al., Langmuir 2012, 28, 3695). We would like to clarify that it is due to our same concern as the reviewer's, we **did not measure** the repulsive forces from nanoparticle motions; instead we modeled and calculated the repulsive forces based on the prisms and the beam-modulated liquid condition changes.

In the revised draft, we made the following changes to accommodate the reviewer's comments:

- (i) in the second paragraph of "Results", where we first discussed nanoparticle diffusion, we acknowledged upfront with a more thorough explanation and citations (included all listed by the reviewer) how NPs diffuse differently within liquid-phase TEM chambers:
"Their motions are slower than that predicted by Stokes-Einstein equation, likely because of an increase in solvent viscosity during imaging or the involvement of nanoparticle-substrate attractions.^{24,37-40} These effects bring the time scale of nanoparticle motions up to the temporal resolution of liquid-phase TEM instrumentation, while as we detail later, they still keep the fundamental nature of internanoparticle interactions involved unchanged."
- (ii) We see it positive that our calculations based on the conventions for nanoparticle self-assembly studies outside TEM correctly predict the physics we observed in our liquid-phase TEM experiment. We added text and references to make it more clear in the "Discussion" section:
"We find our theoretical modeling based on conventions of nanoparticle self-assembly explains our experimental observation, which confirms that, despite the differences of the liquid environment inside a sample chamber from conditions outside the TEM, there appears to be no fundamental difference in internanoparticle interactions."

We also hope to note to the reviewer that thermal energy $k_B T$ can still serve as a useful energy unit in the discussions, given that the slowed motions of nanoparticles are a result of thermodynamic equilibrium (whether due to increased liquid viscosity or due to nanoparticle-substrate interactions). In fact, in the field of micron-scale colloidal self-assembly, researchers often intentionally design colloids to move in a non-Brownian fashion (e.g. with external energy transferred into the particles, see Yan et al., Nature 2011 491, 578; Driscoll et al., Nature Physics 2017, 13, 375) to render unconventional self-assembly structures. In such cases, the convention for energy scale is still $k_B T$.

Reviewer 2 Comment 4: *The authors indicate that only electrostatics and van der Waals control assembly in this case. Several recent reports have suggested the influence of solvation forces are significant in mediating self-assembly (Welch et al., ACS Nano, 10, 2015; Anand et al., Nano Letters, 16, 2016). But the authors have not included these effects in their interaction potential energy model. There*

are standard models for solvation forces, the authors should test these as well in their interaction model.

Reply: We thank the reviewer for the great and insightful comment! In the revised draft, we have performed calculations of hydration force as shown in Supplementary Fig. 10, Table 4 and Note 7. In addition, we commented in the main text that the hydration forces are negligible in our system based on our calculations,

“Regarding hydration interactions that were shown relevant for the orientated attachment of bare metallic nanoparticles during their growth,^{43,52} we follow established forms and calculate that the hydration interactions for a pair of prisms are constantly below $1 k_B T$, negligible in the total free energy responsible for prism attachment configurations (Supplementary Fig. 10, Table 4 and Note 7).”

We would like to note that this result is also consistent with the fact that considering only electrostatic repulsion and van der Waals attraction correctly predicts our experimental observation.

In response to the reviewer’s potential question on a qualitative understanding (besides our calculation results) of why hydration forces are important in the two listed references but not in our work, we list the reasons as follows. These reasons are also included in Supplementary Note 7.

In both listed references (Welch et al. and Anand et al.), hydration forces were found important for naked nanoparticles (not coated with surface ligands) that still grow and coalesce under beam, while we used pre-synthesized nanoparticles coated with charged ligands in our work. As stated in these two and other references, hydration forces concern the water packing structure close to the surfaces. This difference, with and without ligand coating, can have three immediate impacts.

First, extended charged ligands (such as charged thiols in our work) will make the water molecules close to surface much less structured, which have been reported to significantly lower the hydration forces in previous literatures (Refs. 2, 15, 16 in the Supplementary Information).

Second, ligand coated nanoparticles will not be physically as close as naked nanoparticles while as noted in the Anand paper, the decay length of hydration forces is about 1.4 \AA . These two reasons together account for the observation in the Anand paper that is consistent with our work: in their control sample of CTAB-coated nanoparticles, hydration forces are not important in determining the assembly configuration.

Third, charged ligands also introduce additional electrostatic repulsion into the internanoparticle interactions. Electrostatic repulsion is not important in the naked nanoparticle studies in both listed references, while in our case of charged prisms, the electrostatic repulsion stays more significant in varying bond angles than hydration repulsion (also repulsive for the same hydrophilic surfaces), and overwhelms the small magnitude of hydration repulsion (Supplementary Fig. 10). In particular, in the Welch paper, the end-to-end rod attachment was favored due to the much larger hydration repulsion in the side-by-side configuration; while in our work, the tip-to-tip attachment was selected by the long-range electrostatic repulsion, which precludes the side-by-side configuration to even fall into the hydration interaction range.

Reviewer 2 Comment 5: *The results showing bimodal bonding distributions is very similar to the work of Welch et al cited above, and thus again not an entirely new concept. Welch et al also showed two predominant bonding angles for oriented attachment of silver nanorods, which the authors interpreted in terms of differences in solvation shell at various facets on the ends of the nanorods. This result is somewhat similar to what the authors are proposing here, as the bonding depended on facet geometry. I would like to see comparison to this previous article as well, as there are a number of similar aspects between the two studies.*

Reply: We thank the reviewer for the great suggestion! In this listed reference, they measured the probability plot of the attachment angle (defined differently from ours) for two nanorods. For a direct comparison, we put the angle distributions in the Welch paper and in our work as below: the angle distribution of Welch et al. (left) is based on a total of 23 observations, and our bimodal distribution of bond angles (right) shows two clear peaks based on a total of 84 events.

In the Welch paper, they simulated how the activation energy changes as a function of bond angle (Figure 4) at the end of paper. They noted (1) consistent with experiments, most tip-to-tip attachments are at low orientation angle ($\sim 0^\circ$): “Low energy barriers (<15 kT) were predominantly found at rod ends.”, and there are rare observations of high-orientation bond angles ($22 - 43^\circ$) due to a low energy barrier around 30° that lowers hydration forces as rod facets are apart. The authors stated in their work this other

bond angle is a relatively rare event, “for six of 23 attachment events (see Figure 1e), at least one high orientation angle was found. Four of these attachment events were associated with rod-rod angles in the range of $22-43^\circ$, which is, in terms of the SMD model (see Figures 3 and 4), associated with low-energy barriers due to relatively weak solvation forces.”

We appreciate that in the Welch paper, the point of how tip facets fine-tune the interaction energy was brought up. In the revised draft, we cited this paper, made comparisons, and added other references and comments on how the local curvature argument generalizes easily to other nanoparticles. We would like to note that the detailed observation and mechanism for the bond angle distribution in the Welch paper are distinct from ours. In our work, the bimodal distribution is clear, 0° and 60° bond angles are almost equally probable; the high 60° is not a rare event. Our calculation shows 60° angle is due to the morphology detail of local curvature, which plays a crucial role when van der Waals attraction dominates the system. This is the case for nanoparticle self-assembly in aqueous solution coated with ligands, instead of nanoparticle growth studied in the Welch paper (see more details in our response to Reviewer 2 Comment 4). The geometric feature of nanoparticle local curvature generalizes to other nanoparticles.

“In addition to this model system of triangular prisms, previous work on beveled gold prisms⁵³ and gold nanorods⁵² has also reported how small morphology details of nanoparticles determine other interaction types, such as depletion and hydration interactions. In the nanorod paper,⁵² two different assembly configurations were observed and selected by contact-determined hydration interactions.”

Reviewer 2 Comment 6: *There seems to be an error in the second paragraph of page 10, the authors refer to figure 3e, which does not exist.*

Reply: We greatly thank the reviewer for reading our manuscript so carefully and pointing out the error! We meant to refer to Figure 3d. In the revised draft, we corrected this typo and have done more thorough proofreading to avoid other potential typos.

To Reviewer 3

Reviewer 3 considers the work an elegant piece which should be published after some revisions, which we have addressed as listed below.

Reviewer 3 Comment 1: *More generally, one result that is particularly intriguing in my opinion is the fact that the ‘dance’ of nanoparticles in the align and bind case does not seem to follow a purely stochastic process. While the authors carefully described the mechanisms leading to the alignment and attachment of the particles, I would be interested in a more detailed description of what is happening, what are the driving forces, during the last step of the attachment the ‘jump’ from free to attached.*

Reply: We thank the reviewer for reading our manuscript so carefully and for the great suggestion! In the original draft, we summarized the overall driving force at the end of the discussion for Figure 1: electrostatic repulsion determines attachment configuration at a distance while van der Waals attraction facilitates the nanoparticle attachment at short distance. The attachment at short distance due to strong van der Waals attraction can be too fast to be captured gradually by the TEM camera, which is effectively shown as a “jump”. In the revised draft, we revised this description and add more references to make it more clear and better articulated,

“This observation suggests an interesting long-range effect: the tip-to-tip configuration is selected at a distance, before the prisms are in physical contact, and is then successfully attached via van der Waals attraction.”, and “In comparison, tip-to-tip aligned prisms could overcome a relatively low repulsion barrier ($15 k_B T$) to diffuse close and fall into the range of van der Waals attraction (see more details in Supplementary Fig. 7 and Table 2). The last step of attachment driven by this van der Waals attraction is sometimes observed as a jump from free to attached (Fig. 2b). This is likely due to attraction kinetics that are considerably faster than our acquisition capability.⁴³”

Reviewer 3 Comment 2: *Furthermore, in some previous research endeavors (ref 34), the authors used similar nanoplatelets and observed the formation of ‘meta-rods’. Could the authors briefly comment on how the same particles with the same ligand, in very similar conditions self-assembled in such different structures.*

Reply: We thank the reviewer for noticing our previous work and for the great suggestion on clarifying why the observations are different! In our previous work (ref 35, Kim et al., ACS Nano 2016, 10, 9801), we used liquid-phase TEM to directly image **pre-assembled** prisms into 1D face-to-face stacks (meta-rods) in a high salt concentration (0.15 M pH=8 phosphate buffer solution). In other words, the prisms were first assembled outside TEM and we then loaded the assembled prisms into the liquid cell chamber for direct imaging. The procedure in this work is different: the initial state in the liquid cell prior to beam-illumination is individual, non-assembled prisms in DI water and the self-assembly is triggered by radiolysis-generated ions. Then due to prism-substrate interaction, the assembly in our work is quasi-2D and the prisms assemble laterally above the substrate at a much lower ionic strength. In the revised draft, we added notes in the Method section to highlight the above differences.

Reviewer 3 Comment 3: *On page 3, the authors mentioned that the self-assembly is triggered by a combination of change in pH and ionic strength in the cell and they offer in the SI a calculation of the change of both pH and I as function of e- dose. Given that this is the driving force leading to the self-assembly, it would be interesting to either measure those value on chip during the self-assembly process or alternatively to comment on how pH and I change with time during the reaction.*

Reply: We thank the reviewer for the great suggestion! In the original draft, we estimated the effective ionic strength increase and pH change as detailed in Supplementary Fig. 2 and text. In the revised draft, we revised the text to make it more articulated:

“Through radiolysis reactions with water, the imaging beam: (1) monotonically increases the ionic strength, facilitating the counter-ion screening of electrostatics, and (2) increases the acidity, rendering a smaller charge density on prisms (estimated in Supplementary Fig. 2, Notes 1 and 2 following radiolysis equations of water⁴¹). Both changes reach the steady state concentration profiles within seconds upon beam illumination.⁴¹”, and added comments on that these effects do not change the fundamental nature of nanoparticle interactions:

“In addition, we have also shown in our previous work that at this low dose rate range the internanoparticle interactions and self-assembled structures modulated by electron beam can be reproduced by implementing effective ionic strength and pH conditions outside the TEM.³⁵”

If I understood correctly the self-assembly is occurring in the liquid cell in absence of flow and one could expect a build-up of I and pH over time, given that the authors discuss the kinetic of self-assembly, this raises the question on how the self-assembly process is affected by those changing reaction conditions. More simplistically, how does the time-scale of the pH and I change compares to the time-scale of self-assembly?

Reply: We thank the reviewer for the great question! Yes, the self-assembly occurs in absence of flow. The time-scale of pH and ionic strength change is within seconds after beam illumination, while the time scale of assembly is 10-30 mins when most prisms are involved in the chaining process. In addition, the pH and ionic strength change is achieved by reaching a steady state based on the radiolysis reaction network of the solvent, which will stay the same under the constant dose rate. In the revised draft, we added additional discussions to clarify the above:

“Both changes reach the steady state concentration profiles within seconds upon beam illumination.⁴¹”.

Reviewer 3 Comment 4: *On page 5, the authors discussed the fact that the prisms acted as divalent species and referred to fig S4 for more details. While the statistics in the SI indicated that in the linear assemblies the divalent species are favored compared to the trivalent species, this is not such a clear case for the cyclic chains. To get a better feel for the real fraction of divalent to trivalent species, it would be interesting to know what are the fraction of linear to cyclic chains in the sample, and more over what is the overall fractions of monovalent, divalent, trivalent and multivalent species in the entire system and more over how these fractions are changing when the conversion increases. How does branching vary with conversion?*

Reply: We thank the reviewer for the great suggestions! The linear and cyclic chains are formed in different beam conditions, and, as a result, in two different ionic strengths. In the original draft, we discussed the final bonding geometry difference in prisms in Supplementary Fig. 4, where the prisms in cyclic chains have more number of connections (more branching) than that in the linear chains.

In the revised draft, we added the branching – time curves for both the linear chain and the cyclic chain as Supplementary Fig. 14 and added discussions to articulate the difference:

“This increased number of bonds (valency) leads to an assembled network woven by prism nodes, i.e. branched prisms, and cyclization events (Supplementary Fig. 4). The evolution of the fraction of branched prisms over time shows an obvious increase in comparison with the constant trend for linear chains (Supplementary Fig. 14).”.

Reviewer 3 Comment 5: *On page 5, the authors claimed that the self-assembly follows a step-growth kinetic. It would be interesting to see this claim supported not only with X_n vs t , but also with the change in polydispersity over time.*

Reply: We thank the reviewer for the great suggestion! In the original draft, we showed two supporting measurements of step-growth polymerization: $\bar{X}_n = 4[M]_0kt + 1$ (Fig. 1e) as the reviewer mentioned, and Flory-Schulz distribution $n_x/N_L = (1-p)p^{x-1}$ (Fig. 1f, Supplementary Fig. 5). We followed the reviewer's suggestion and verified that polydispersity index (PDI) also follows the predicted $\text{PDI} = 2 - 1/\bar{X}_n$. The fitting graph is shown on the right.

In the revised draft, we included the graph as Supplementary Fig. 5c, and added details in the corresponding Supplementary Note 4 and discussions in the main text:

“The polydispersity index (PDI) of the chains also fits well with the relation of $\text{PDI} = 2 - \frac{1}{\bar{X}_n}$, which is expected for step-growth polymerization of divalent monomers (Supplementary Fig. 5c and Note 4).”

Reviewer 3 Comment 6: *On page 8, the authors discussed how certain bond angle are more favored than others (fig2). In the supporting information the authors also both measure the experimental distribution of bond angle (fig S12) and develop a model to calculate the energy associated with each configuration (Fig S11). Could the authors directly compared the ‘theoretical’ results with the experimental results, to get a better sense of how accurate the phenomenological description given of this complex self-assembly mechanism is accurate?*

Reply: We thank the reviewer for the great suggestions and completely agree with the reviewer that it is good to have a side-by-side comparison between experimental and calculation results of the bond angles. We thank the reviewer for reading the supplementary figures so carefully! The theoretical model predicts a transition of preferred assembly from 0° bond angle at an average curvature of 0.2 nm^{-1} to 60° bond angle at and below a curvature of 0.05 nm^{-1} . This prediction is confirmed by the experimental observation of assembly configurations and supports our definition of flat and rounded tips. All the calculations (Supplementary Figs. 12 and 13) are combined and shown as a 3D graph in Figure 4d, and we explained using this graph, “As shown in Fig. 4d, the interaction energy valley propagates continuously from a stable bond angle of 0° for two round-tipped prisms (top black line) to a stable bond angle of 60° for one round-tipped and one flat-tipped prisms (bottom black line), which agrees with the experimentally observed two bonding motifs (Supplementary Figs. 11-13).”

Reviewer 3 Comment 7: *On page 10, the authors discussed the effect of the tip curvature on the more favored self-assembled structure. Could the authors comments on how the change in the prism shape might influence the crystalline facet exposed and how this could lead to different grafting density of the ligand on the surface which in turn might influence the self-assembly process.*

Reply: We thank the reviewer for this great comment! We agree with the reviewer that the ligands might bind with a different density on these facets, but as shown in this work, we did not need to include this effect in our calculation to reproduce the results. This is probably due to the following three reasons: (i) van der Waals attraction between organic ligands is negligible; (ii) the ligand density is low anyway at the prism tip; and (iii) at low ionic strength, the electrostatic repulsion is long-ranged and the charged ligands on the basal facets dominated the interaction. In the revised draft, we added additional details as follows:

“Note that the ligand density of truncated tips may be different from that of rounded tips;^{50,51} we show here that we do not need to include this potential difference in our interaction calculation to reproduce the bimodal bond angle distribution. This is likely due to the negligible van der Waals attraction

component between organic ligands and the dominant electrostatic repulsion from the ligands on the large basal planes.”

Reviewer 3 Comment 8: *On page 13, the authors give some experimental details. However, it is not enough to enable other to reproduce these experiments. They do not report what is the final concentration of the gold nanoplatelets suspension, nor do they describe with enough details the ligand exchange procedure and cleaning process used to make sure that there is no remaining CTAB in solution.*

Reply: We thank the reviewer for the great suggestion! In the original draft, we included experimental and computational details in the supplementary notes but not in the “Method” section of the main text. In the revised version, we tried to detail as much as possible both the experimental and calculational procedures. For one, after the thiol modification, the extinction peak value of the prism suspension is ~ 2 (prism concentration = ~ 50 pM). The CTAB-thiol ligand exchange was done with water washing such that the residue CTAB is estimated to be as low as ~ 5 μ M.

Reviewer 3 Comment 9: *On page 13, the authors described the model used for their calculation. Furthermore, some parameter used in the calculations are given in the SI. However, it would be interesting to see a more detailed description of the modelistic approach used maybe in SI.*

Reply: We thank the reviewer for the great suggestion! In the revised draft, we included all the calculation parameters used and added more text in the Method section and additional supplementary figures/tables in the supporting information.

Reviewer 3 Comment 10: *Finally, on a cosmetic note, the authors seem to favor (fig 1d, 1f, 2f) a color scheme based on the use of light green and slightly darker green, the difference is not obvious.*

Reply: We thank the reviewer for the great suggestion! Given that Fig. 1 is already quite color-intensive and the other two reviewers did not comment on this aspect, we decided to keep the color scheme the same but added additional descriptions on the symbol shapes in the figure legend to make it more clear.

REVIEWERS' COMMENTS:

Reviewer #2 (Remarks to the Author):

The authors have sufficiently addressed all of my comments and I therefore recommend the manuscript is published as is.

Reviewer #3 (Remarks to the Author):

In this revised version of the manuscript, the authors took into account the comments from the first round of revision. The manuscript feels now more complete.

I think that the manuscript could be published in its present form.